# Nonlinear scaling of resource allocation in sensory bottlenecks

**Laura R. Edmondson**[1,3], **Alejandro Jiménez-Rodriguez**[2,3], **Hannes P. Saal**[1,3]
[1]Department of Psychology
[2]Department of Computer Science
[3]Sheffield Robotics
The University of Sheffield
{lredmondson1,a.jimenez-rodriguez,h.saal}@sheffield.ac.uk

## Abstract

In many sensory systems, information transmission is constrained by a bottleneck, where the number of output neurons is vastly smaller than the number of input neurons. Efficient coding theory predicts that in these scenarios the brain should allocate its limited resources by removing redundant information. Previous work has typically assumed that receptors are uniformly distributed across the sensory sheet, when in reality these vary in density, often by an order of magnitude. How, then, should the brain efficiently allocate output neurons when the density of input neurons is nonuniform? Here, we show analytically and numerically that resource allocation scales nonlinearly in efficient coding models that maximize information transfer, when inputs arise from separate regions with different receptor densities. Importantly, the proportion of output neurons allocated to a given input region changes depending on the width of the bottleneck, and thus cannot be predicted from input density or region size alone. Narrow bottlenecks favor magnification of high density input regions, while wider bottlenecks often cause contraction. Our results demonstrate that both expansion and contraction of sensory input regions can arise in efficient coding models and that the final allocation crucially depends on the neural resources made available.

## 1 Introduction

In biological sensory systems, information transmission is often constrained by a neural bottleneck, where the number of output neurons is vastly smaller than the number of input neurons. For example, there are many more photoreceptors in the retina than there are retinal ganglion cells in the optic nerve. Sensory bottlenecks force compression of information [20], and their presence and narrowness affects the layout of receptive fields [17]. Efficient coding theory has been used to predict how the brain should allocate its limited resources in these scenarios by removing redundant information [1–4].

Prior work has typically assumed that the density of input receptors is constant [1, 6]. However, in biological sensory systems, receptors are often not distributed uniformly across the sensory sheet, but vary in their density. In vision, the density of cones in the retina differs by an order of magnitude between the fovea and the periphery [9, 22]. In touch, mechanoreceptors are much more densely packed in the fingertips than they are in the palm [14].

How, then, should the brain efficiently allocate output neurons when the density of input neurons is nonuniform and a sensory bottleneck constrains the total number of output neurons (see Fig. 1A for an illustration)? A plausible solution might simply prescribe a constant ratio of input to output neurons, and therefore preserve proportional allocation, independent of the width of the bottleneck. However,

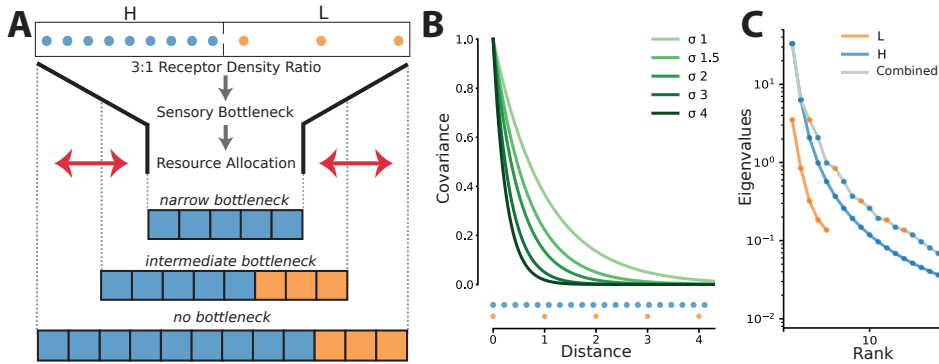

Figure 1: **A.** Illustration of the resource allocation problem. Sensory inputs from two regions with different receptor densities ($H$ and $L$) pass through a neural bottleneck. At a given bottleneck size, how many output neurons should be dedicated to inputs from each of the two input regions? The optimal allocation might depend on the width of the bottleneck. **B.** Sensory inputs are correlated according to a covariance function (here, negative exponential) that decays with distance between receptors on the sensory sheet. Note that this function is evaluated at different distances $|x_i - x_j|$ depending on the density of sensory receptors. Two potential receptor densities are indicated at the bottom of the panel in blue ($H$) and orange ($L$). The covariance function is plotted with different decay constants $\sigma$. **C.** In efficient coding models that maximize decorrelation of sensory inputs, the resource allocation problem can be solved by principal component analysis on the sensory inputs of each region individually and then sorting the combined set of eigenvalues; the region each successive eigenvalue in the combined set originated from determines where each additional output neuron's receptive field will fall (see main text for details).

sensory signals arising from densely packed receptors are more correlated than those from sparsely distributed receptors, suggesting diminishing information gain from allocating outputs neurons to high density over low density input regions. Hence, denser regions should be under-represented in the bottleneck, relative to their input density. Finally, a case can also be made for expansion of denser input regions, as this ensures the increased spatial resolution afforded by densely packed receptors can be fully taken advantage of in subsequent processing stages.

Which of these three ideas is correct? Here, we answer this question analytically and in numerical simulations, and demonstrate that both expansion and contraction of sensory input regions can be optimal in efficient coding models. We show that the final allocation depends on the width of the bottleneck and the precise nature of spatial correlations in the sensory inputs.

## 2 Background: Decorrelation/Whitening

We focus on linear second-order models that maximize information through decorrelation of sensory inputs. Decorrelation has been proposed as an important principle at lower levels of sensory processing [10], where sensory bottlenecks appear most prevalent. This approach is equivalent to minimizing the mean-squared reconstruction error in the noiseless case. We will argue that our results extend to (some) more complex models in section 6.1.

Mathematically, our goal is to determine the $m \times n$-dimensional weight matrix $\boldsymbol{W}$ that decorrelates the $n$-dimensional sensory inputs. Correlations in the inputs arise because nearby receptors respond similarly to sensory stimuli; this relationship weakens as the distance between receptors increases (Fig. 1B). A sensory bottleneck is introduced by restricting ourselves to $m < n$ outputs. Sensory inputs are represented by the zero-mean $n \times z$ matrix $\boldsymbol{X}$, containing $z$ $n$-dimensional sensory inputs. The whitened data $\boldsymbol{W}\boldsymbol{X}$ should be uncorrelated, such that

$$\boldsymbol{X}^T\boldsymbol{W}^T\boldsymbol{W}\boldsymbol{X} = \boldsymbol{I}. \tag{1}$$

This can be achieved by setting $\boldsymbol{W} = \boldsymbol{\Sigma}^{-\frac{1}{2}}$, where $\boldsymbol{\Sigma} = \boldsymbol{X}^T\boldsymbol{X}$. Solutions are in the form

$$\boldsymbol{W} = \boldsymbol{P}\boldsymbol{\Lambda}^{-\frac{1}{2}}\boldsymbol{U}^T, \tag{2}$$

where $\boldsymbol{\Lambda}$ is a diagonal matrix containing the eigenvalues of $\boldsymbol{\Sigma}$ and $\boldsymbol{U}$ contains its eigenvectors. Whitening filters are not unique [15], and any orthogonal matrix $\boldsymbol{P}$ will yield equally valid whitening filters. A popular solution in cases without a bottleneck ($m = n$) that yields localized filters (receptive fields) is known as ZCA (Zero-Phase Component Analysis) [5] and sets $\boldsymbol{P} = \boldsymbol{U}$.

In cases with a sensory bottleneck ($m < n$), a solution can be found by solving an orthogonal procrustes problem [6, 7]:

$$\boldsymbol{P}^* = \min_{\boldsymbol{P}} \left\| \boldsymbol{W}_{opt} - \boldsymbol{P}\boldsymbol{\Lambda}^{-\frac{1}{2}}\boldsymbol{U}^T \right\|_F^2, \tag{3}$$

where $\|\cdot\|_F$ denotes the Frobenius norm. Here, $\boldsymbol{W}_{opt}$ is an $m \times n$ matrix containing idealized local receptive fields [see 6, for strategies to set its values]. Setting $\boldsymbol{W}_{opt}$ to the identity matrix in the no-bottleneck case will recover the ZCA solution described earlier. $\boldsymbol{\Lambda}$ ($m \times m$) and $\boldsymbol{U}$ ($n \times m$) are as above, but retain only the $m$ components with the highest associated eigenvalues, thereby projecting the sensory data $\boldsymbol{X}$ onto the space spanned by its principal components (PCA). $\boldsymbol{P}$ is an $m \times m$ orthogonal matrix.

# 3 Derivation

## 3.1 Whitening of two input regions

We assume that input regions with different densities are not bordering each other, such that the covariance between any pair of receptors from different regions will be zero. We have tested numerically that this provides a valid approximation in the case of two regions directly bordering each other and only introduces marginal error (see Supplemental Materials). Under this assumption, the covariance matrix will be a block diagonal matrix. In the specific case of two regions $H$ (high receptor density) and $L$ (low receptor density), $\boldsymbol{\Sigma}$ therefore breaks down as follows:

$$\boldsymbol{\Sigma} = \begin{bmatrix} \boldsymbol{\Sigma}_H & \mathbf{0} \\ \mathbf{0} & \boldsymbol{\Sigma}_L \end{bmatrix} \tag{4}$$

It can be shown by application of the Cauchy Interlacing Theorem that the block diagonal matrix structure of $\boldsymbol{\Sigma}$ implies that its eigenvalues are identical to the combined set of the eigenvalues of $\boldsymbol{\Sigma_H}$ and $\boldsymbol{\Sigma_L}$ [16]; similarly, $\boldsymbol{U}$ is a block matrix that can be reconstructed from $\boldsymbol{U}_H$ and $\boldsymbol{U}_L$:

$$\boldsymbol{\Lambda} = \begin{bmatrix} \boldsymbol{\Lambda}_H & \mathbf{0} \\ \mathbf{0} & \boldsymbol{\Lambda}_L \end{bmatrix} \quad \text{and} \quad \boldsymbol{U} = \begin{bmatrix} \boldsymbol{U}_H & \mathbf{0} \\ \mathbf{0} & \boldsymbol{U}_L \end{bmatrix} \tag{5}$$

For a sensory bottleneck with $m$ output neurons, we retain only the $m$ largest eigenvalues from $\boldsymbol{\Lambda}$ along with their associated components in $\boldsymbol{U}$. We can now see that this is equivalent to sorting the eigenvalues from both regions and finding the $m$ largest ones in the combined set (see Figure 1C for a visual example). Eigenvalues chosen from region $H$ imply that the receptive field of the added output neuron also falls onto region $H$[1]. Thus, the problem of how output neurons are allocated to either input region is solved by calculating and sorting the eigenvalues associated with each individual input region. In the following, we will show how these calculations can be solved analytically for exponential covariance functions. In section 5, we will discuss an example where the eigenvalues are calculated and sorted numerically for an empirically determined covariance function.

## 3.2 Calculation of eigenvalues

In the following, we will restrict ourselves to one-dimensional inputs only (see section 6 for a discussion of the 2D case). We assume that the covariance decays exponentially with receptor distance. The elements of the covariance matrix are then calculated as follows (see Fig. 1B):

$$\boldsymbol{\Sigma}_{ij} = \exp(-\sigma|x_i - x_j|), \tag{6}$$

where $x_i$ is the location of the $i$th receptor and $\sigma$ is the decay constant. For exponential covariance functions, it is convenient to express the eigenvalue-eigenvector problem in a continuous domain. In this case, the eigenvalues can be calculated analytically using the following integral homogeneous equation:

$$\lambda_k \phi_k(x) = \int_a^b \exp\left(-\gamma(x)|x-y|\right)\phi_k(y)dy, \tag{7}$$

where $\phi_k(x)$ is the $k$th eigenfunction and $\lambda_k$ its corresponding eigenvalue. The domain length (region size) is set as $S = b - a$. When seen as a discretization of the continuous version, the PCA of the sampled data amounts to a expansion/compression of one of the regions in the spatial domain. As a consequence, the rate of decay becomes $\gamma(x) = \sigma$ when $x$ is on the low density region, and $\gamma(x) = r\sigma$ otherwise, with $r$ denoting the ratio of high versus low density.

By using the Fourier transform, it can be shown that equation 7 is equivalent to the law for the classical harmonic oscillator in each region (see Supplemental Materials for proof):

$$-\frac{d^2}{dx^2}\phi_k(x) = \left(\frac{\gamma}{\lambda_k} - \gamma^2\right)\phi_k(x) = \mu_k\phi_k(x). \tag{8}$$

Equation 8 denotes the Laplacian eigenvalue problem which, for a finite spatial domain of length $S$ has the following known solution:

$$\phi_k(x) = \begin{cases} \sqrt{\frac{2}{S}}\cos(\sqrt{\mu_k}x), & k \text{ odd} \\ \sqrt{\frac{2}{S}}\sin(\sqrt{\mu_k}x), & k \text{ even} \end{cases} \tag{9}$$

In the case where spatial correlations don't extend over the full sensory sheet, i.e. when the region size is big relative to the spatial extent of the correlations, as is usually the case in the sensory systems, it can be shown that the Dirichlet initial conditions $\phi(a) = \phi(b) = 0$ hold, yielding the the corresponding eigenvalues $\mu_k = \frac{k^2\pi^2}{S^2}$ (see Supplemental Materials for proof and full derivation of the exact boundary conditions). Finally, the PCA and the Laplacian eigenvalue problems share the same eigenfunctions and their eigenvalues are related by the equation $\lambda_k = \frac{2\sigma}{\mu_k + \sigma^2}$.

For two regions $H$ and $L$, we can therefore calculate their eigenvalues as:

$$\text{Region } H: \lambda_h = \frac{2\sigma}{h^2\pi^2 S^{-2} + \sigma^2}, \quad \text{and Region } L: \lambda_l = \frac{2\sigma}{l^2\pi^2 S^{-2}r + \sigma^2 r}, \tag{10}$$

where $r > 1$ is the ratio of higher and lower densities, and $h, l \in \mathbb{N}$ denote successive eigenvalues for regions $H$ and $L$, respectively.

### 3.3 Allocation in the bottleneck

To calculate how many output neurons are allocated to region $H$ as a function of the number of neurons allocated to region $L$, we set $\lambda_h = \lambda_l$ and substitute equation 10. This yields

$$h = \frac{\sqrt{l^2\pi^2 r + S^2\sigma^2 r - S^2\sigma^2}}{\pi}. \tag{11}$$

It becomes apparent that for $l = 1$, i.e. the first neuron allocated to region $L$, we have already assigned $h = \frac{\sqrt{(r-1)S^2\sigma^2 + r\pi^2}}{\pi} > \sqrt{r}$ neurons to region $H$. As we allocate more neurons to region $L$, the ratio $\frac{h}{l}$ simplifies to: $\lim_{l\to\infty} \frac{h}{l} = \sqrt{r}$. The fraction of neurons allocated to each region therefore depends on the size of the bottleneck and converges to $\frac{\sqrt{r}}{1+\sqrt{r}}$ and $\frac{1}{1+\sqrt{r}}$ for $H$ and $L$ respectively. Note that this result is independent of the region size $S$.

Extending our results to more than two regions is straightforward, but requires substituting equation 10 by an analogous system of equations, whose solutions define the relationships between eigenvalues from all regions.

## 4 Results

We calculated the predicted allocation of output neurons for different decay constants $\sigma$ and density ratios $r$ over all possible bottlenecks widths. An illustrative example is shown in Fig. 2A and B: the

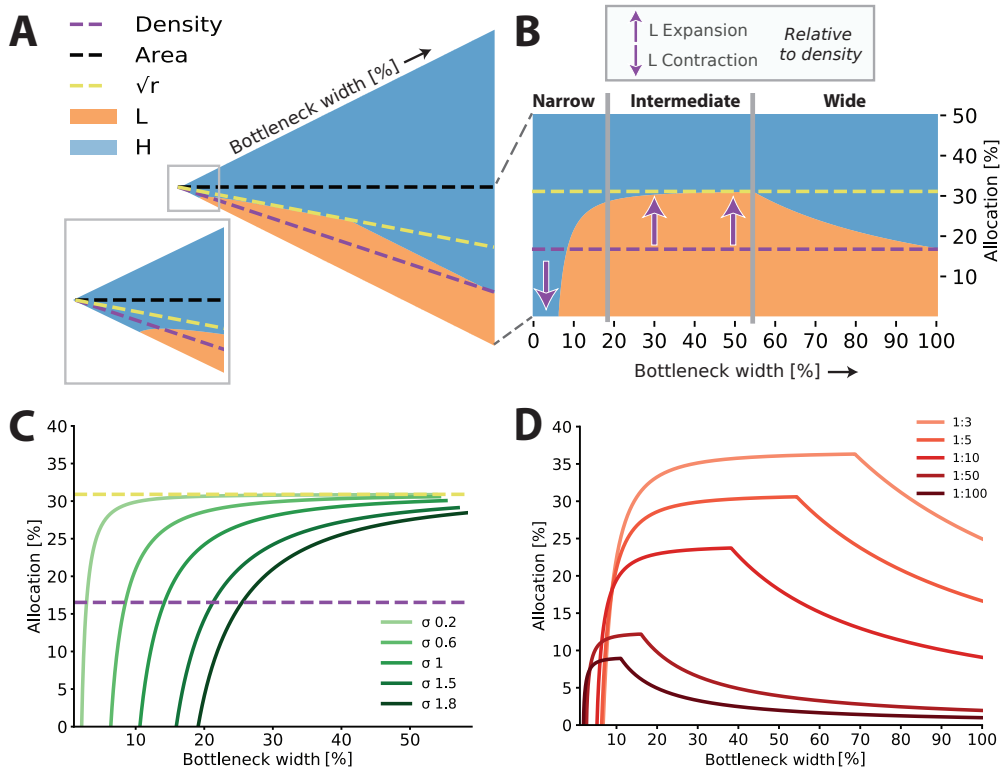

Figure 2: **A.** Allocation of output neurons to the high density (blue) or low density (orange) input regions for different bottleneck widths for an example with density ratio $r = 5$, a decay constant $\sigma = 0.6$, and a region size $S = 300$. Vertical slices through the expanding triangle denote different bottleneck widths. Dashed lines indicate allocation according to region size (black, kept the same for both regions in our analysis), receptor density (purple) and a mathematically derived asymptote (yellow, see main text for details). The inset depicts a zoomed-in version of the allocation for extremely narrow bottlenecks. **B.** Allocation for same parameters as in A, but normalized to the number of output neurons for each bottleneck width. Different proportional allocations are obtained for narrow, intermediate, and wide bottlenecks, leading to expansion or contraction of input regions in the bottleneck relative to their receptor density. **C.** Allocation boundaries for different decay constants $\sigma$ as function of bottleneck width for a density ratio $r = 5$. More spatially restricted (faster decaying) covariance functions lead to a contraction of low-density input regions. Dashed lines as in A. Note that the plot cuts off before the allocation boundary decays back to the density ratio shown in B. **D.** Allocation boundaries for different input density ratios at $\sigma = 0.6$. For intermediate bottlenecks, all curves tend towards a limit determined by the ratio. Higher ratios cause the low-density region to saturate at narrower bottlenecks (one-to-one mapping of receptors to outputs), after which the allocation decays back to the density ratio.

allocation of output neurons is a nonlinear function of the bottleneck width, both in absolute (Fig. 2A) and relative number of units (Fig. 2B). Specifically, different allocation strategies are apparent for narrow, intermediate, and wide bottlenecks as follows:

1. For *narrow* bottlenecks, all or most of the output neurons are allocated to the high density input region, leading to an expansion of this region in the bottleneck relative to its share of receptors. Conversely, the low density input region is contracted and might not even be represented at all in extremely narrow bottlenecks. Both the extent of expansion/contraction and the range of bottleneck widths over which it occurs is affected by the decay constant $\sigma$: larger decay constants, i.e. more spatially localized correlations, increase the amount of expansion of the high density input region, which is represented exclusively for extremely

narrow bottlenecks (see Fig. 2C). The density ratio $r$ also affects the initial expansion, but to a lesser extent (Fig. 2D).

2. For *intermediate* bottlenecks, output neurons are allocated at a ratio of $\sqrt{r}$ to the high density over the low density region. In this regime, the high density region will contract relative to its receptor input density (cf. yellow dashed lines in Figs. 2A,B, and C). We note that this asymptote does not depend on the decay constant $\sigma$, but the decay affects how fast the allocation converges to this ratio. While the allocation is driven towards the limit as the bottleneck widens, it might not be reached in practice, if the spatial extent of the input correlations is low (see dark green line in Fig. 2C).

3. Finally, for *wide* bottlenecks the low density region will reach the point where each receptor is assigned an individual output neuron, and therefore all information from this region is captured in the bottleneck. In our method, this corresponds to exhausting the number of eigenvalues arising from this input region (see Fig. 1C for a visual example). All additional outputs neurons will therefore be allocated to the high density region. This is apparent in the figures as a slow decay of the allocation boundary to the input density ratio. In the full case (no bottleneck), input and output densities are matched.

The allocation of output neurons in the bottleneck directly affects the spatial resolution with which stimuli can be resolved on the sensory sheet. Adding neurons increases the spatial frequency of the associated eigenvectors (cf. eq. 9): higher frequencies support smaller receptive fields and therefore increased spatial resolution. Dedicating output neurons to a given input region will therefore trade off accuracy increases in this region at the expense of the other region. Our results suggest that narrow bottlenecks favor increased spatial resolution mainly in high density regions to the detriment of the lower density region; at wider bottlenecks the differences in spatial resolution between the two regions even out, and are indeed smaller than the difference in input densities alone would predict.

In summary, efficient coding schemes support both expansion and contraction of receptor inputs in the bottleneck; a crucial factor in the resulting allocation is the overall width of the bottleneck itself.

# 5 Empirical example: natural image statistics

So far, we restricted ourselves to exponential covariance functions. How do our results translate to other spatial relationships? Natural images induce spatial correlation that follow a different decay function: the power spectral density (and therefore the distribution of eigenvalues) of natural scenes follows a well-known power law, where power decreases with $1/f^2$ for increasing spatial frequencies $f$ [11]. We tested numerically how neurons in a visual bottleneck should be allocated to different input regions, reflecting the fact that the density of cone photoreceptors is not constant across the retina.

## 5.1 Methods

We calculated the covariance between pairs of pixels from a set of natural images. As in our previous analysis, we restricted ourselves to the 1D case. We included 2,000 randomly sampled images from the SALICON image data set[2] [13], converted the images to 8-bit grayscale, and then extracted luminance values along horizontal lines extending 160 pixels each. Images were $480 \times 640$ pixels in size yielding 1,920 samples per image, and therefore 3.8m samples in total. The resulting covariance function decays smoothly with pixel distance, as expected, but induces more far-ranging correlations that an exponential decay (see inset in Fig. 3A). We again restricted our analysis to comparing two input regions, testing out different receptor density ratios. For the high density region we directly assigned neighboring image pixels to receptors. For the low density regions, we calculated covariances at larger pixel distances as specified by the respective density ratios, $r = 2, 5, 10$. Next, we calculated the eigenvalues of the covariance matrices for the high density and the three low density regions. As expected, the eigenvalue spectrum followed a power law (linear relationship on a log-log plot, see Fig. 3B). Finally, we sorted the empirical eigenvalues from high and low density regions to determine the proportion of output neurons with receptive fields falling onto the high and low density input regions, respectively, as described in section 3.

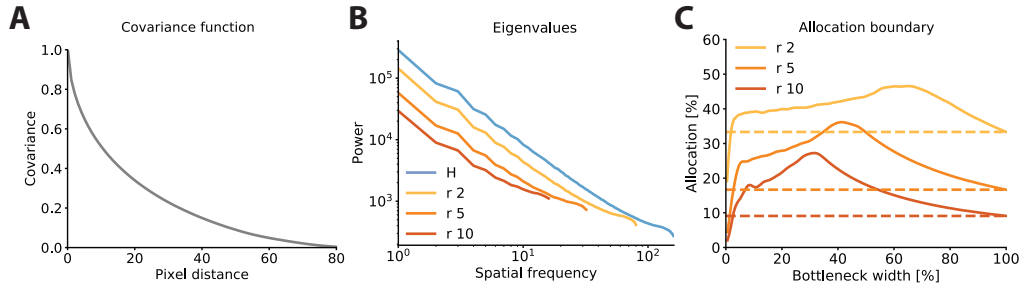

Figure 3: Empirical results on natural image data set. **A.** Covariance between pairs of pixels as a function of distance. The covariance decays fast initially, but wide-ranging dependencies can be observed. **B.** Eigenvalue spectrum for different receptor densities. For the high density region (blue), we included every pixel in the original images. For low density regions, we sampled every 2nd, 5th, and 10th pixel, respectively (orange-shaded lines). As expected, the eigenvalues follow a power law. **C.** Allocation boundaries for different density ratios. The area below the boundary denotes allocation to the low density input region, while the area above is allocated to the high density region. Dashed lines show allocation proportional to receptor density. Both expansion and contraction of the high-density region can be observed.

## 5.2 Results

We found that the covariance structure imposed by natural images resulted in expansion of high density inputs for narrow bottleneck widths (Fig. 3C). Indeed, extremely narrow bottlenecks lead to an exclusive representation of the high density region, ignoring inputs from the low density region entirely. Conversely, wider bottlenecks lead to a contraction of the high-density region. The bottleneck imposed by the optic nerve is very narrow and inputs from the fovea are over-represented in the optic nerve; as such, our findings are at least qualitatively in line with these experimental findings. Still, our results in this section are not intended to make precise predictions about the allocation of fibers in the optic nerve: we did not model the filtering properties of the lens, which blurs visual inputs in the peripheral retina, and we did not calculate our results in 2D (see section 6 for further discussion) or take the difference in size between the fovea and the retinal periphery into account. Instead, our results are meant to highlight that resource allocation under an efficient coding model is shaped not just by the width of the bottleneck, but also by the precise nature of the correlations between individual receptors.

## 6 Discussion

We have shown that efficient coding models nonlinearly scale their resource allocation in sensory bottlenecks under nonuniform input densities. That is, the limited number of outputs neurons is not simply allocated proportional to receptor density. Rather, input regions might expand or contract in the bottleneck and the main driver behind this effect is the width of the bottleneck itself: narrow bottlenecks cause over-representation, while wider bottlenecks favor under-representation of high-density inputs. The extent of spatial correlations across the sensory sheet also influences the results, as does the range of receptor densities in the different regions.

### 6.1 Implications for efficient coding models

Our results emphasize that the presence of sensory bottlenecks can have important consequences for the resulting neural representations. Many standard efficient coding models assume that the number of input and output neurons is matched, or that the pool of output neurons is virtually unlimited, though recent work has started to explore the effect of bottlenecks on neural coding in more detail [17].

Nonuniform allocation of output neurons is a hallmark of efficient coding models: neurons should be allocated proportional to the probability of each stimulus, such that more likely stimuli are encoded

with higher accuracy [see e.g. 8, for a recent model]. While this principle appears straightforward, in practice it can lead to complex and sometimes counter-intuitive effects on neural allocation [19] and its perceptual consequences [21]. In contrast to prior work, here we focused on resource allocation in the presence of nonuniform *receptor density*, while assuming a spatially uniform stimulus probability distribution with respect to where on the sensory sheet a stimulus might fall. While such an assumption might be warranted for the visual system, in other sensory systems the spatial distribution of stimuli can be highly non-uniform. For example in the tactile system, we are much more likely to come into contact with an object on our fingertips than on any other region of our hand. Interestingly, the density of mechanoreceptors is also much higher on the fingertips than anywhere else on the hand. Indeed, nonuniform receptor placement might be a way for evolution to bias the resulting neural representations towards ecological or behaviourally relevant priors.

Our results demonstrate that even the simplest and most commonly employed efficient coding model (linear, noiseless, second-order) yields an interesting and surprising relationship between the resulting allocation and the bottleneck width. The presence of this relationship is therefore not dependent on noise or nonlinearities. While our results were derived with classical ZCA-style whitening [5] in mind, they are also valid for other variants, as long as these project the sensory inputs into the lower-dimensional space spanned by the principal components of the sensory inputs. This includes models that optimize for stimulus reconstruction accuracy and take into account sensory noise [6], alongside decorrelation. Indeed, higher-order models such as independent component analysis (ICA) also include this step in the undercomplete case, i.e. when a bottleneck is present [12]. Our results therefore hold for this class of models as well. Furthermore, employing a simple model means that resource allocation can be solved analytically under our cost function. This paves the way for future analyses of more complex models, for example introducing nonlinearities by means of kernel PCA.

## 6.2 Resource allocation in biological sensory systems

Based on our findings, we make two specific predictions for resource allocation in biological sensory systems. First, the width of the bottleneck determines which input regions will expand or contract their representation in the bottleneck. This could be tested by comparing sensory systems, for example the visual pathway, across a number of species with different bottlenecks. Second, the precise nature of the correlation function determines whether the resulting representation favors contraction or expansion of high-density input regions. For example, the covariance function induced by visual stimuli (cf. section 5) caused different levels of expansion and contraction than the exponential covariance function. This suggests observable differences in the resulting representations across different senses, even in cases when their bottleneck widths might be similar.

Here, our results were limited to one-dimensional receptor surfaces. We expect similar principles to apply for two-dimensional sensory sheets, such as the retina in vision, and the skin in touch. However, two-dimensional surfaces that are tiled by receptive fields of different sizes scale differently than one-dimensional ones [18], and this aspect would need to be taken into account.

## 6.3 Applications and future work

Bottlenecks are common in machine learning models to help with generalization and have recently attracted renewed interest in the field of deep neural networks [17]. Non-uniform inputs have not been studied in detail, however appear useful for robotics applications, particularly where power and size constraints are important. In both cases, our work suggests that the size of the bottleneck is critically important in shaping the resulting representations.

Future work might extend our approach to multiple receptor populations: both touch and vision rely on multiple receptor classes that occur at different absolute densities, but also exhibit different density gradients across the sensory sheet. Vision relies on three different types of cones as well as rods, while tactile feedback includes responses from at least four classes of mechanoreceptors in non-hairy skin. In both systems individual receptor classes exhibit different but overlapping tuning functions, implying that the responses from different receptor classes will be correlated. Our results suggest that these correlations should impact resource allocation.

**Acknowledgements**

We would like to thank Mark Humphries for comments on an earlier version of this manuscript. This work was supported by the Wellcome Trust [209998/Z/17/Z] and by the EU Horizon 2020 program as part of the Human Brain Project [HBP-SGA2, 785907].

## Footnotes

[1]Localized receptive fields in $\boldsymbol{W}$ can only be obtained if $\boldsymbol{P}$ is a block matrix and $\boldsymbol{W}_{opt}$ places output units according to the breakdown of the eigenvalues. A shortcut to calculate localized receptive fields with minimal extent is to set $\boldsymbol{P} = \boldsymbol{U}$ and the calculate $\boldsymbol{W}$ using QR decomposition. Note that our method can also accommodate non-localized receptive fields, if we take the accuracy with which inputs on the sensory sheet can be resolved as a proxy for resource allocation (see further discussion in section 4): retaining additional principal components from $\boldsymbol{U}$ will increase spatial resolution selectively for the region the eigenvector originated from.

[2]The full data set can be downloaded from http://salicon.net.

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
