[Supplementary Material]

# Supplemental Materials for:
# Nonlinear scaling of resource allocation in sensory bottlenecks

**Laura R. Edmondson**[1,3]**, Alejandro Jiménez-Rodriguez**[2,3]**, Hannes P. Saal**[1,3]
[1]Department of Psychology
[2]Department of Computer Science
[3]Sheffield Robotics
The University of Sheffield
{lredmondson1,a.jimenez-rodriguez,h.saal}@sheffield.ac.uk

## 1   Derivation of precise expressions for the eigenvalues and eigenfunctions

For an exponential covariance matrix, the PCA problem can be approached using the continuum equivalent (functional PCA):

$$\lambda_k \phi_k(x) = \int_a^b \exp\left(-\gamma(x)|x-y|\right)\phi_k(y)dy, \tag{1}$$

which is a homogeneous Fredholm integral equation. In this section we assume $\gamma(x) = \sigma$, that is a single region with the same covariance function between receptors. We now proceed to find the eigenfunctions and eigenvalues. Note that a solution of the discrete version:

$$\lambda_k \phi_k(x_i) \approx h \sum_{i=a}^b \exp\left(-\sigma|x_i-y_j|\right)\phi_k(y_j) \tag{2}$$

will approach that of the continuum problem as $h \to \infty$, where $i = a, \ldots, b$ is a given partition of the corresponding domain.

**Theorem 1.** *The solutions of (1) are eigenfunctions of the Laplacian operator with eigenvalues* $\mu_k = \frac{2\sigma}{\lambda_k} - \sigma^2$. *In one dimension, solutions* $\phi_k$ *satisfy:*

$$-\frac{d^2}{dx^2}\phi_k(x) = \mu_k \phi_k(x). \tag{3}$$

*Proof.* Equation (1) is a convolution. In the limit of an infinite domain, we can apply the Fourier transform on both sides,

$$\lambda \mathcal{F}(u)(\eta) = \mathcal{F}(W * u)(\eta) = \mathcal{F}(W)\mathcal{F}(u)(\eta), \tag{4}$$

where $W(x,y) = \exp\left(\sigma|x-y|\right)$, to obtain

$$\lambda \hat{U}(\eta) = \frac{2\sigma}{\eta^2 + \sigma^2}\hat{U}(\eta),$$

and, reorganizing the terms

$$-(\eta i)^2 \hat{U}(\eta) + \sigma^2 \hat{U}(\eta) = 2\sigma \hat{U}(\eta)/\lambda. \tag{5}$$

Applying the inverse Fourier transform, we obtain the desired result.

$$-\frac{d^2 u}{dx^2} = \left(\frac{2\sigma}{\lambda} - \sigma^2\right) u$$

This result can be also obtained by direct differentiation of (1) twice. $\qquad\square$

## 1.1 Region size

For a given region size $S$, the eigenvalues of the second derivative are given by:

$$\phi_k(x) = \begin{cases} \sqrt{\frac{2}{S}} \cos(\sqrt{\mu_k}x), & k \text{ odd} \\ \sqrt{\frac{2}{S}} \sin(\sqrt{\mu_k}x), & k \text{ even} \end{cases}, \tag{6}$$

The traditional Dirichlet conditions $u(0) = u(S) = 0$, give the Laplacian eigenvalue $\mu_k = \frac{k^2\pi^2}{L^2}$. However, the integral equation (1) introduces an unconventional boundary condition that we derive in the following.

The boundary condition implies that a given interval $[a, b]$, as used in the main text, should be mapped to an appropriate interval in which the boundary conditions are satisfied. This mapping gives the exact value of $\mu_k$ for the integral equation.

**Theorem 2.** *The eigenfunctions $\phi_k$ of the Laplacian must satisfy the following boundary condition on the interval $[a, b]$*

$$\phi'(a) = \sigma\phi(a) \tag{7}$$
$$\phi'(b) = -\sigma\phi(b), \tag{8}$$

*and the corresponding eigenvalues $\mu_k$ can then be calculated from the solutions $\eta = \sqrt{\mu_k}$ to the following transcendental equations:*

$$\eta a = \arctan\left(\frac{-\sigma}{\eta}\right) - k\pi \tag{9}$$

*for even $k$, and*

$$\eta a = \arctan\left(\frac{\eta}{\sigma}\right) - k\pi \tag{10}$$

*otherwise.*

*Proof.* We start by finding constraints for the solutions $\phi_k(x)$ at the extremes of the interval. First, replacing $x = a$, we find that it should satisfy

$$\phi_k(a) = \frac{1}{\lambda} \int_a^b \exp\left(\sigma(a - y)\right) \phi_k(y) dy. \tag{11}$$

In turn, the first derivative of (1)

$$\phi'(x) = -\frac{\sigma}{\lambda} \left[ \int_a^x \exp\left(-\sigma(x - y)\right) \phi_k(y) dy - \int_x^b \exp\left(\sigma(x - y)\right) \phi_k(y) dy \right],$$

becomes

$$\phi'(a) = \frac{\sigma}{\lambda} \int_a^b \exp\left(\sigma(a - y)\right) \phi_k(y) dy \tag{12}$$

at $x = a$, yielding the following functional equation

$$\phi'(a) = \sigma\phi(a) \tag{13}$$

Replacing $\phi_k$ with the solutions in (6) yields:

$$-\eta \tan\left(\eta a\right) = \sigma, \tag{14}$$

where we have replaced $\sqrt{\mu_k}$ by $\eta$. Reorganizing, we obtain the desired transcendental equation whose solutions will give the correct values of $\sqrt{\mu_k}$. $\qquad\square$

## 1.2 Correct interval for the eigenfunctions

The previous result allows us to find the right interval to plot the eigenfunctions found using equation 6 when comparing them with the simulation results. Indeed, given a domain length $S$, the limits of integration in (1) are given by:

$$a_k = \frac{1}{\sqrt{\mu_k}} \arctan\left(-\frac{\sigma}{\sqrt{\mu_k}}\right) \tag{15}$$

$$b_k = \frac{1}{\sqrt{\mu_k}} \arctan\left(\frac{\sigma}{\sqrt{\mu_k}}\right), \tag{16}$$

for odd $k$ and

$$a_k = \frac{1}{\sqrt{\mu_k}} \arctan\left(-\frac{\sqrt{\mu_k}}{\sigma}\right) \tag{17}$$

$$b_k = \frac{1}{\sqrt{\mu_k}} \arctan\left(\frac{\sqrt{\mu_k}}{\sigma}\right), \tag{18}$$

for even $k$.

## 1.3 Large region length and covariance limits of the eigenvalues

The eigenvalues from theorem 2 differ from the ones found from the Dirichlet conditions $\phi(a) = \phi(b) = 0$, which, as we use them in the main text, have the form

$$\mu_k = \frac{\pi^2 k^2}{S^2}, \tag{19}$$

where $S$ is the region size. However, it can be shown analytically and by means of simulations that the former tend to the used expression under two asymptotic limits.

Figure 1: Eigenvalues calculated from theory and numerical evaluation for $\sigma = 0.5$, and $n = 300$ points in the interval $[-150, 150]$

First, observe that as $\sigma \to \infty$, equation (14) implies $\eta a = \frac{\pi}{2}$, and therefore for narrow covariance matrices $\eta = \sqrt{\mu_k} \approx \frac{\pi k}{2S}$. Second, note that as the length $S \to \infty$, the limits in (15) also approach

infinity where solutions to (1) also satisfy $\phi(-\infty) = \phi(\infty) = 0$. Additionally, eigenvalues of the form (19) satisfy equation (14) in the large region size limit.

We conclude that for large enough regions or narrow covariance matrices, the traditional Laplace eigenvalues constitute a good approximation. These conditions imply that the spatial extent of the correlations should be small, considering the full population of receptors. Expressed differently, typical stimuli should be much smaller than the size of the sensory sheet. This will generally be the case in sensory systems, where only a small group of receptors would respond to a given stimulus. As shown in Figure 1 there is close agreement between this approximation and a numerical evaluation for a typical example case as discussed in the main paper.

However, if the regions considered are small enough such that most receptors would be correlated, the results derived above should be taken into account.

## 2 Non-block covariance matrices

In the main text, we assume that the input regions $H$ and $L$ are independent and do not border each other, allowing an analytical solution using block matrices. Therefore, covariances for all pairs of receptors from the different regions will be 0. Do our results apply when we allow for positive cross-covariances? We tested this by solving equation (3) in the main text numerically for different density ratios ($r = 2, 5, 10$) and decay constants ($\sigma = 0.1, 2, 4$) for both the block and non-block variants. This allowed us to directly calculate localized receptive fields. We allocated neurons to the input regions according to where the peak of its localised receptive field fell. The resulting allocations (expressed in percentages) were compared for the block and non-block variants. Comparisons were only made for bottleneck widths where both regions still had units left to be allocated.

We found that the allocation for non-block covariance matrices closely tracked those derived for block matrices. The average error across all runs and bottleneck widths was 0.31 percentage points, a marginal difference. The maximal error observed was 5 percentage points, although we noticed that errors larger than 2 percentage points were only observed for $\sigma = 0.1$, i.e. extremely wide covariances, and generally at narrower bottleneck widths where the allocation boundary was changing rapidly. Examining the receptive fields, we noticed that the only observable difference between the block and non-block variants was the inclusion of receptive fields that straddled the border between both regions $H$ and $L$ in the non-block case i.e. there are a small number of neurons whose receptive fields cover both regions. We conclude that the differences between the block and non-block variants are negligible and our approach provides a valid approximation.