[Reviews · NeurIPS 2019]

Reviewer 1



Update after rebuttal ---------------------------- Thanks the the authors for their response. I have read their comments, but have decided to maintain my original score, as my concerns about the input noise and correlations largely remain. Original review -------------------- I enjoyed reading this paper. The paper leverages the fact that for a block diagonal input covariance matrix (which occurs if the inputs to the two regions are independent), then the optimal solution relies on simply taking the top $m$ eigenvalues from the jointly sorted list--this means that the output space can have proportionally different representations of the input, compared to the original ratio (called $r$ in the paper). This is a simple, yet surprising, fact that is clearly presented. The mathematical details are also easy to follow. I am unaware of other work in the efficient coding literature that addresses this problem in the same way, as such, I believe the work is original. The figures are clean and elegant and also greatly aid the presentation. I would have liked to see the authors address a couple of additional scenarios in their work: - What if there are not just two different receptor densities, but three, four, or many? In that case, what does the resource allocation look like. This is mentioned in the discussion, but it seems like a simple enough extension that may be worth including in this paper. - What about input noise? Decorrelation is optimal only when there is no input noise. The presence of (independent) input noise means that the system may want to average over the redundancies in the inputs to improve the signal to noise ratio. Can this analysis be extended to this scenario (c.f. Doi et al, ref. [6] in the paper). - I think more should be said about the block diagonal assumption. It seems unlikely that nearby receptors (fovea vs periphery, or fingertip vs hand) will experience totally independent inputs (there are strong long range correlations in natural stimuli). I appreciate that the assumption allows for an analytic solution, but perhaps more can be said about how the analytic solution breaks down as this assumption is violated. - There is a difference between receptor densities that tile *different* parts of the input space (e.g. cones in the fovea vs periphery, or mechanoreceptors in the fingertip vs the rest of the hand), vs receptors that tile the *same* part of the input space (e.g. overlapping cell types in the retina, or multiple mechanoreceptor types in the skin). The paper seems to address the former scenario, where correlations between the receptors would be reduced, but what about the latter? Perhaps the authors can comment in the discussion on whether this analysis is appropriate for overlapping receptor populations that each tile the same input space.

Reviewer 2



Major: What is the explicit objective function: reconstruction, information preservation? In what noise regime are we? input noise, output noise, large, infinitesimal? My understanding is that the objective function is information preservation in the infinitesimal noise regime for both input and output. The objective function should be stated explicitly, and the equations of the article should ideally be derived from this initial objective function. Minor "In vision, the density of cones in the retina differs by several orders of magnitude between the fovea and the periphery" => closer to one order of magnitude: see https://www.ncbi.nlm.nih.gov/pmc/articles/PMC4985666/ eq7: define a, gamma, x. where does the number of receptors appear? eq8: how did you apply the Fourier transform? steps missing. eq10: what is the unit of a? what is the number of receptors? Do you assume that both regions have the same size (low density and high density regions)? Possible discussion point: why are receptors arranged in an inhomogenous way in the first place? clarity: OK quality: good. originality: haven’t seen anything like this. significance: medium/high. implications for AI?

Reviewer 3



The paper considers an efficient coding setup in which the input receptors are non-uniformly distributed. The optimal neuronal allocation is derived based on varying sensory bottleneck and input density. To my knowledge, this particular setup has not been carefully studied in the previous work. Overall the study is well executed and the logic is clear. The method used is also technically sound. The paper is generally well written, and relatively easy to follow. I have two main concerns: First, the current model setup ignores many critical ingredients, such as noise in the output, noise in the input, output nonlinearity, and the metabolic cost. It is known that these factors can fundamentally change the optimal coding configuration. The model presented currently is a linear model without output (spiking) noise and metabolic cost. It is difficult to judge how these additional ingredient might change the solution derived in the present work. This is an important issue because it affects the interpretation and the potential contribution of this work, in terms of how well the theory could explain the neurophysiology, as well as whether the regime considered in the paper is a relevant regime for understanding the physiology. Second, while the theoretical derivations are neat and the analytical results are interesting, the connections to experiments are rather weak, at least as it stands now. The application to the natural images statistics with the 1D model is helpful, but the relevance to the physiology is largely unclear. This makes the significance of the work uncertain. More specific comments: Concerning modeling the non-uniform input- Does input X contain any input noise? If so, how does the non-uniform input density affect the characteristics of the input noise? If not, how would adding input noise change the results? It is not obvious how receptor density is modeled in Eq (6). It would useful to clarify. Line 26- it would useful to cite previous work to support the claim here. Line 188,189- it would be useful to cite the relevant experimental literature here. The title doesn’t seem to fully capture the gist of the paper; I’d suggest something like ”Optimal resource allocation in sensory bottlenecks with non-uniform input sampling”. Of course, this is entirely up to the authors.. ** edit after the rebuttal I found the authors' response and the discussion with other reviewers to be helpful. Although I still like to see more ingredients such as input/output noise, metabolic constraints, nonlinearity, to be added to the theory to make it more relevant for neurophysiology, at the same time, I think the current version is already interesting enough. Although I am not increasing my score here, I'd like to vote for acceptance of this paper.

[Author Response · NeurIPS 2019]

We would like to thank the reviewers for their positive and constructive comments. We address the major points below.

**Cost function**   Reviewers asked to clarify the cost function we used, and whether/how changing the cost function would alter the results. We agree that the cost function should be clearly specified in the text and we will do so in the final version. Here, we did not include an explicit noise model; our result can be derived by either minimizing the reconstruction error (under mean-squared error) or by maximizing information, which restricted to second-order statistics equates to maximizing explained variance. We agree with the reviewers that it would be very interesting to explore alternative cost functions. We chose our approach for the following reasons:

1. We wanted to demonstrate that even the simplest and most commonly employed model yields an interesting and (to us) surprising relationship between the resulting allocation and the bottleneck width. The presence of this relationship is therefore not dependent on noise or nonlinearities.

2. Often, more complex models include a projection to PC space as the first step, and thereby yield the same allocation described in the paper. For example, this applies to a common variant of ICA (fastica) in the undercomplete case, which we have confirmed numerically. Similarly, the approach taken by Doi & Lewicki (2014) that includes an explicit noise model contains the same projection. Of course, alternative formulations of the noise model or cost function that would affect allocation directly are conceivable.

3. Finally, resource allocation can be solved analytically under our cost function. This paves the way for future analyses of more complex models, for example introducing nonlinearities by means of kernel PCA.

**Generalization of results**   First, regions of different sizes can be incorporated by adjusting the model's length scale, $a$. Second, the solution can be extended to $n > 2$ regions: eigenvalues are calculated for each region independently, and can be sorted (and interlace) irrespective of the number of regions. Third, the 2D case can also be solved by a straightforward adaptation of the Laplacian eigenvalue problem. We will clarify these points in the final version. Finally, we have confirmed numerically that the block-matrix approach provides a valid approximation (see paper appendix).

Figure 1: Resource allocation in the optic nerve calculated from natural image statistics.

**Implications for neuroscience**   Several comments asked about the match between our analysis of natural image statistics and neurophysiological data. We initially included this example simply to demonstrate the importance of the covariance function on resource allocation. The original analysis was based on equal region sizes and therefore precluded a direct comparison with data from the visual literature, as the fovea is much smaller than the retinal periphery. We have now re-expressed our results to take this size difference into account. Importantly, the data set and methods are the same as before, the only difference being the number of inputs for each of the regions to account for the difference in size. We took the central 5 degrees of the retina as the fovea, yielding a size ratio of 1:166. We assumed 260,000 cones in the fovea, and the rest in the periphery, containing around 5,940,000 cones (values extrapolated from Curcio, 1990; Wells-Gray et al., 2016).

The resulting allocation boundary is shown in Fig. 1. Importantly, if we use this result to calculate the implied ratio between cones and (midget) retinal ganglion cells (RGCs) in the optic nerve, we find a predicted ratio for the fovea of just over 1, indicating that each RGC should receive input from only 1 to 2 cones; the same ratio has been well established experimentally. For the periphery, we predict on average inputs from 10-20 cones onto a single RGC, again in agreement with the literature.

As the density of cones varies smoothly across the retina, future work should build on these results to explore the principles of RGC allocation on a finer scale; however, such more complex analysis is beyond the scope of the current paper.

**Implications for machine learning**   Bottlenecks are common in ML models to help with generalization and have recently attracted renewed interest in the field of deep neural networks. Non-uniform inputs have not been studied in detail, however appear useful for robotics applications, where power and size constraints are important. In both cases, our work suggests that the size of the bottleneck is critically important in shaping the resulting representations.

[Meta-Review · NeurIPS 2019]

The reviewers agree that this paper has interesting contributions in both theory and application with proposing a novel model for resource allocation in biological sensory systems. The authors have addressed most of the reviewers concerns, however, it would be great if authors explicitly state the noise assumptions in the final revision.